# Comprehensive Evaluation of Appreciation of *Rhododendron* Based on Analytic Hierarchy Process

**DOI:** 10.3390/plants13040558

**Published:** 2024-02-19

**Authors:** Jincheng Liang, Yaoli Chen, Xuexiao Tang, Yin Lu, Jinghui Yu, Zongbo Wang, Zetian Zhang, Hao Ji, Yu Li, Purui Wu, Yue Liu, Ling Wang, Chuanhuang Huang, Bizhu He, Wei Lin, Lijin Guo

**Affiliations:** 1Key Laboratory of Genetics and Germplasm Innovation of Tropical Special Forest Trees and Ornamental Plant, Ministry of Education, College of Tropical Agriculture and Forestry, Hainan University, Haikou 570228, China; glj13302302912@163.com (J.L.); 21220954000003@hainanu.edu (Y.C.); t1667837170@163.com (X.T.); 18275541190@163.com (P.W.); wygzyx06@163.com (Y.L.); 2International Magnesium Institute, College of Resources and Environment, Fujian Agriculture and Forestry University, Fuzhou 350002, China; luyin0417@163.com (Y.L.); wang13837956853@163.com (Z.W.); zhangzetian0808@163.com (Z.Z.); 17516383972@163.com (H.J.); 3College of Plant protection, Fujian Agriculture and Forestry University, Fuzhou 350002, China; yjh2685132589@163.com; 4College of Forestry, Fujian Agriculture and Forestry University, Fuzhou 350002, China; 17740109663@163.com; 5Fuzhou Qinting Lake Park, Fuzhou 350012, China; qthgyglc@163.com; 6Fuzhou Botanical Garden, Fuzhou 350021, China; 13859094296@163.com; 7College of Horticulture, Fujian Agriculture and Forestry University, Fuzhou 350002, China; 15108954085@163.com

**Keywords:** *Rhododendron*, analytic hierarchy process, comprehensive evaluation, ornamentality, adaptability

## Abstract

Qinting Lake Park has effectively imported *Rhododendron* varieties from Zhejiang Province. The analytic hierarchy process was employed to devise an evaluation framework to evaluate the ornamental and adaptive features of these species. Subsequently, we conducted a standardized evaluation of 24 species for their ornamental and adaptive traits under controlled cultivation conditions. The findings indicated that the percentage of ornamental flowers in the first-level index was significantly greater than the other two factors, indicating that the ornamental value of flowers was the most important in the evaluation of *Rhododendron* ornamental value. Among the secondary indicators, the proportion of flower color and flower weight was significantly higher than that of other factors, which had the greatest impact on the evaluation results. The 24 *Rhododendron* species were classified into two grades based on their ornamental value, as determined by index weights and scoring standards. *Rhododendron* ‘Xueqing’, *Rhododendron* ‘Big Qinglian’, and *Rhododendron* ‘Jinyang No. 9’ exhibited superior ornamental value and demonstrated more favorable suitability for garden applications.

## 1. Introduction

There exist approximately 1025 *Rhododendron* L. species globally, categorized into eight subgenera [1]. These species have a wide distribution across Europe, Asia, and North America [2]. *Rhododendron*s are highly esteemed in horticulture due to their exceptional ornamental qualities, including various flower types and vibrant colors. More than 600 species of *Rhododendron* have been cultivated and introduced worldwide. Natural hybridization and easy hybridization in cultivation have developed numerous hybrids with superior ornamental value compared to their wild counterparts [3]. As a result, it is cultivated worldwide as an essential ornamental plant [1,4].

China possesses a vast wealth of *Rhododendron* germplasm resources, with over 600 species [5]. Since the publication of FRPS, numerous new taxa have been continuously discovered [6,7,8,9]. *Rhododendron* cultivars are highly valued garden plants in China due to their graceful shape, vibrant flowers, and delightful fragrance, making them one of the most famous flowers in the country [10]. They are widely used internationally for landscaping and greening, as well as for culinary and medicinal purposes [11]. Most wild *Rhododendron*s grow in sparsely vegetated mountainous areas or beneath pine forests at 500–1200 m elevations. They are widely recognized as typical indicators of acidic soils, thriving in humid and cold regions with diffused light, and displaying intolerance to heat. *Rhododendron*s grown in low-altitude or low-latitude areas with higher temperatures struggle to adapt. They are prone to stunted growth, species degradation, or even death, significantly limiting the broad application of more *Rhododendron* species in garden landscapes [11,12]. Fujian experiences prolonged high temperatures, sometimes reaching 40 °C, during summers and autumns. This climatic condition makes *Rhododendron*s in the region susceptible to leaf yellowing and desiccation, greatly hindering the widespread cultivation and utilization of *Rhododendron* in Fujian. Therefore, it is significant to study the ornamental properties of *Rhododendron* cultivars and their adaptability for introduction and cultivation in Fujian to identify high-quality germplasm resources suitable for the local area.

There is yet to be a globally recognized and feasible method to assess *Rhododendron*’s adaptability and aesthetic qualities comprehensively. Various evaluation methods have been used, such as hierarchical analysis, principal component analysis, and the gray correlation method. Hierarchical analysis, also known as the Analytic Hierarchy Process (AHP), was introduced by Saaty T.L. in the early 1970s and offered unique advantages [13]. It breaks down decision-making elements into multiple levels, allowing qualitative and quantitative analysis to address the problem systematically [14]. AHP determines the importance ranking through weighting analysis, providing a scientific and feasible optimal solution. AHP has been increasingly used, in recent years, to evaluate the ornamental value of garden plants [15,16,17,18], including in the maples [15], peonies [4,18], begonias [19], and other among the ornamental plants. However, a more comprehensive evaluation of new *Rhododendron* species’ ornamental and adaptive qualities is needed. Conducting a multi-factor and multi-criteria quantitative evaluation of garden ornamental plants presents challenges due to the various factors influencing their ornamental value. Therefore, it is necessary to establish an objective, comprehensive, scientific, and quantitative evaluation system.

Furthermore, existing studies on applying AHP in tree evaluation have focused more on the ornamental properties of plants and overlook their adaptive ability after introduction and planting. This hinders the introduction and domestication of high-quality ornamental plants suitable for the local area. Considering the adaptability of plants to planting is essential for creating long-term ornamental landscapes. A study was conducted in Qinting Lake Park, Fuzhou, to evaluate and analyze 24 *Rhododendron* species from Zhejiang province. The study aimed to establish a preliminary model for evaluating the aesthetic and adaptability of *Rhododendron*s and screen for cultivars with elegant posture, strong drought resistance, and suitability for planting in Fuzhou. These selected *Rhododendron*s can significantly enhance the quality of the garden landscape and serve as a reference for future *Rhododendron* introductions in Qinting Lake Park and Fujian Province. By planting and screening these *Rhododendron*s, a beautiful and sustainable garden can be created, attracting visitors worldwide and providing resources for garden plant landscaping.

## 2. Results

### 2.1. Evaluation and Ranking of Rhododendron Cultivars

To assess Rhododendrons’ ornamental and biological characteristics, we distributed questionnaires to teachers and students (*n* = 35) specializing in relevant fields [20]. Using a five-point system, we scored 15 evaluation criteria outlined in Table 1. Then, we derived a comprehensive assessment and analytical table for the 24 selected Rhododendrons from Qinting Lake Park, integrating the evaluation criteria to derive a ranking. Based on the rankings, the 24 Rhododendron variants were divided into two tiers: Level I (3.1 > score > 2.9) with 15 species suitable for garden application and Level II (2.5–2.9) with nine species representing Rhododendron variants with standard comprehensive utility, as shown in Table 2.

### 2.2. Cluster Analysis

A cluster analysis used the scores derived from 15 characteristic indexes across 24 *Rhododendron* cultivars. The pedigree chart depicting these clusters is illustrated in Figure 1. Upon review of the pedigree diagram, distinct groupings become apparent: the first cluster, comprising ‘Xueqing’, ‘Changchun No. 4’, ‘Shengchun No. 1’, ‘Yuanyangjin’, ‘Chunbo’, ‘Big Qinglian’, ‘Shengchun No. 2’, ‘Jinyang No. 9’, ‘Yudataohua’, ‘Hongyang’, ‘Xiaotaohong’, ‘Chuixiao’, ‘Changchun No. 2’, ‘Shengchun No. 4’, ‘Huanjing’, and ‘Kunlun’ jade’, appear to form a cohesive cluster. The second cluster, including ‘Wanzi Qianhong’, ‘Shanlu’, and ‘Ruixue’ are observed to cluster together as a distinct group. ‘White Changchun’, ‘Yinhong Chunjuan’, and ‘Fengmei No. 1’ also exhibit a unique clustering pattern; however, the cluster’s boundaries are less defined, suggesting a more inconclusive association among these cultivars. This revised analysis highlights three major clusters among the *Rhododendron* varieties, emphasizing the distinctive characteristics and relationships within and between these groups, with the third cluster requiring further investigation to determine its coherence.

### 2.3. Heat Map Analysis

This revised analysis highlights three major clusters among the *Rhododendron* varieties, emphasizing the distinctive characteristics and relationships within and between these groups. The heat map results (refer to Figure 2) indicate that among the 24 *Rhododendron* varieties, specific characteristics scored notably higher, while others showed lower ratings: strong performances were observed in petals (C3), flower diameter (C4), flower display (C6), and inflorescence (C7) across the majority of cultivars. However, there were lower scores for flower type (C1), flower color (C2), flower quantity (C5), young leaf color (C10), and growth potential (C14), indicating areas for potential improvement. Notably, ‘Wanzi Qianhong’, ‘Shengchun No.1’, ‘White Changchun’, and ‘Shanlu’ exhibited higher scores in flower-related attributes. However, these cultivars demonstrated lower scores in growth potential (C14) and adaptability (C15). Conversely, ‘Xueqing’, ‘Big Qinglian’, and ‘Jinyang No. 9’ showcased greater adaptability than their counterparts. Additionally, ‘Wanzi Qianhong’, ‘Light Makeup’, ‘Ruixue’, and ‘Illusion’ demonstrated higher ratings in plant type compared to other varieties. This revised analysis provides a more precise depiction of the strengths and weaknesses across different *Rhododendron* cultivars, emphasizing specific areas for improvement and the standout characteristics of each variety.

### 2.4. Principal Component Analysis (PCA)

It can be seen from Figure 3 that the first sorting axis explains 27.7% of the variables, and the second sorting axis explains 21.5% of the variables. Variables such as flower type (C1), petal (C3), flower number (C5), and flower display (C6) showed significant correlations with both the first and second ordination axes.

## 3. Discussion

### 3.1. Analysis of the Key Factors Influencing Ornamental Value

Flower color and quantity are pivotal traits in ornamental plants [21,22,23]. Among the factors influencing flower ornamental properties, flower color (C2) and flower quantity (C5) are the primary limiting factors, accounting for weights greater than 1. Within flower traits, color (C2) and quantity (C5) emerge as primary limiting factors, each with weights greater than 1. Additionally, flower display (C6), flower length (C9), and flower type (C1) present weight values exceeding 0.7, serving as secondary limiting factors. These findings corroborate the results reported by Wang et al. (2023) [24]. This is because the flower color and quantity of *Rhododendron* have the most significant visual impact on tourists, attracting their attention and appreciation, which aligns with the comprehensive evaluation results of *Rhododendron* cultivars conducted by Ye et al. (2020) [25]. When applying *Rhododendron* in garden settings, the primary considerations should be given to flower color, flower quantity, flower display, flower length, flower type, and plant adaptability, consistent with the evaluation conclusions reached by Ye et al. (2020) [25] for 30 *Rhododendron* cultivars. Secondary considerations encompass nine factors, including leaf color and plant type, to avoid indiscriminate applications and ensure robust adaptability and heightened ornamental appeal.

Furthermore, a comprehensive evaluation using the hierarchical analysis model revealed that there are 15 *Rhododendron* species at level I (with a rating between 3.1 and 2.9) regarding their suitability for garden application. On the other hand, nine species are classified at level II (with a rating between 2.5 and 2.9), indicating moderate suitability for garden use. This is due to the introduction of 24 *Rhododendron* varieties in Qin Ting Lake Park, some of which have specific issues. For example, the level I variety ‘Shengchun No. 2’ is considered average regarding ornamental value and adaptability. Similarly, the level II variety ‘Yinhong Chunjuan’ has inherent defects and requires better adaptation for ornamental purposes or garden greening. Based on these findings, minimizing the use of the nine level II *Rhododendron* varieties is advisable when introducing *Rhododendron* in the Fuzhou area. Instead, selecting level I varieties will ensure better adaptation to local climatic conditions and create a more pleasing ornamental effect. Additionally, these results can serve as a reference for the cultivation and introduction of other *Rhododendron* varieties.

### 3.2. PCA for Selecting Optimal Rhododendron Cultivars

The cluster analysis identified ‘Xueqing’, ‘Big Qinglian’, and ‘Jinyang No. 9’ as cultivars with superior adaptability and ornamental value. The 24 *Rhododendron* cultivars were classified into three groups. Examination of the heat map revealed favorable scores for traits such as petal characteristics (C3), flower diameter (C4), flower display (C6), and inflorescence (C7). Conversely, lower scores were observed for flower type (C1), flower color (C2), flower quantity (C5), young leaf color (C10), and growth potential (C14). Integrating these findings with the cluster analysis suggests potential similarities among the four *Rhododendron* cultivars, contributing to their relatively robust adaptability. PCA was employed to validate the experimental results further. The results strongly indicate that the most influential factors are petal diameter (C4), followed closely by flower display (C6), flower type (C1), and flower color (C3). This contradicts the primary factors identified by AHP, highlighting a discrepancy in determining key factors. The AHP method employed various evaluation indicators from domestic and international sources, reducing subjective bias and enhancing outcome reliability. However, both PCA and AHP have inherent limitations. AHP may introduce bias due to inadequate expert scale data or the use of rectangular judgments, undermining objectivity. PCA’s reliance on sample data may compromise the exclusivity of indicator selection. Potential synergies between PCA and AHP should be explored to enhance the efficacy of AHP. Constructing a comprehensive and scientifically robust evaluation framework specifically for *Rhododendron*s becomes imperative. PCA is pivotal in dimensionality reduction, favoring indicators with higher variance contribution rates to maintain robust data integrity. These selected indicators, categorized by correlation, can integrate with PCA to capture interrelationships, resulting in a multi-tiered target system. Ultimately, a matrix serves to consolidate a singular comprehensive evaluation. PCA was used to analyze the data to further verify the experimental results.

### 3.3. Study Limitations

However, this study has certain limitations. The absence of molecular biological or chemical characteristic analysis is a notable limitation, as such an analysis could have provided a deeper understanding of the genetic and biochemical basis of the ornamental traits in *Rhododendron* species. Though not covered in our current research, this comprehensive approach is essential for fully understanding these species.

Another limitation is the primary focus on the ornamental evaluation of *Rhododendron* species using the AHP. While this methodology is suitable for assessing aesthetic attributes, it needs to fully address the adaptability of these species to various environmental conditions, which is crucial for their successful cultivation and viability.

Our research methodology is highly applicable to the unique climatic conditions of China’s southeastern coastal regions, including Fujian Province, Zhejiang Province, and others. These areas experience a typical subtropical climate characterized by high temperatures, abundant rainfall, and high humidity during the summer. These climatic conditions significantly impact the development of ornamental traits in *Rhododendron* species. We have prioritized selecting *Rhododendron* varieties with similar native climatic characteristics to those of the southeastern coast, ensuring their adaptability. Our experiments have validated this approach, as most *Rhododendron* varieties have demonstrated good adaptability. Considering the economic development and demand for landscape plants in these coastal areas, our approach is practical and market-relevant.

Despite its limitations, our study introduces a new method for evaluating the ornamental characteristics of plants in horticultural and landscape planning. We provide valuable insights for selecting ornamental plants in climates and environments similar to those of China’s southeastern coastal regions, contributing to enhancing garden aesthetics and design.

Moving forward, we plan to incorporate a molecular biological or chemical characteristic analysis to understand *Rhododendron* species better. We also intend to include adaptability assessments in our evaluation criteria to ensure a holistic approach considering aesthetic qualities and environmental resilience.

## 4. Materials and Methods

### 4.1. Overview of the Study Area

Qinting Lake Park (26°124029′ N, 119°308261′ E) is located in Fuzhou, Fujian Province, situated within the subtropical monsoon climate zone, characterized by an annual average temperature ranging between 18 to 22 °C. The park receives an average annual sunshine duration of 1700 to 1980 h, with the coldest month averaging temperatures of 6 to 10 °C and the hottest month reaching 33 to 37 °C [21]. The park’s layout, circularly designed around the lake, is significantly influenced by the central lake, affecting air humidity and enriching soil nutrients with its water. The park’s vegetative layout primarily consists of sparse forests and grasslands, integrating *Rhododendron*s beneath the forest canopy, fostering a naturally semi-shaded environment. Since 2013, Qinting Lake Park has introduced 74 species of *Rhododendron* cultivars from Jinhua City, Zhejiang Province, including ‘Chunbo’, ‘White Changchun’, ‘Big Qinglian’, ‘Taiwan Daye’, ‘Dan-Zhuang’, and ‘Shengchun No. 2’, etc., which have been cultivated on a pilot basis on the south bank of the park to create a *Rhododendron*-dominated landscape feature garden.

### 4.2. Plant Material and Cultivation

This study, building on the evaluation of landscape plants both domestically and internationally [20,22,26] and the ornamental effects observed from existing introductions in the garden, selected 24 *Rhododendron* species (Table 3) for comprehensive analysis, focusing on their potential application in gardens. These species underwent one year of unified cultivation management to observe and record their growth, development status, and adaptability. It is important to note that prior extensive research by the authors has already assessed the adaptability issues of *Rhododendron* species [27,28,29]. Therefore, the species chosen for this study had already adapted well to the experimental environment. Additionally, to better showcase the morphological characteristics of the selected *Rhododendrons*, detailed descriptions of their specific shapes have been provided in the Appendix A.

Cultivation management involves optimizing the soil structure by replacing the original matrix with a mix of loess, peat soil, and organic fertilizer at 4:1:1 (*v*:*v*:*v*). Water and fertilizer management adjusts according to precipitation. Spring and summer involve morning and afternoon spray irrigation, with timely drainage during rainstorms to prevent root rot. During *Rhododendron*’s bud stage, increase sprinkler irrigation for flowering needs and reduce it in autumn and winter. Fertilization follows the ‘thin fertilizer application’ principle, adding phosphorus and potassium before flowering and nitrogen after. Pest control includes spraying an 80% dichlorvos solution or 40% omethoate every 7–10 days after April’s withering flowers to eliminate overwintering adults and prevent etiolation and insect pests [24].

### 4.3. Comprehensive Evaluation Analysis Model

#### 4.3.1. Establishment of the Model

Adopting AHP for system analysis requires first analyzing the goals and nature of the problem that needs to be achieved, decomposing the problem into different constituent factors, and then combining these factors together according to different affiliations, initially establishing a multilevel analytical model, and then transforming the problem into an order of importance of the various factors concerning the goal level [30]. Specifically, AHP is a multi-indicator judgment method that decomposes a complex problem into different indicators and groups indicators according to their affiliation and constructs an ordered recursive hierarchy. Two-by-two comparative judgments are made on the indicators to rank the importance of the indicators [20]. AHP can be applied in many fields, such as manufacturing, engineering, and environmental science [20].

Based on the garden ornamental nature of *Rhododendron* [31], as well as referring to the suggestions of related professionals, 15 evaluation indicators of *Rhododendron* related to gardening were screened out, and a comprehensive evaluation and analysis model was preliminarily established based on these 15 indicators (Figure 4). The model consists of three parts: target layer A, constraint layer B, and criterion layer C [25]. Objective layer A is the comprehensive evaluation and analysis of *Rhododendrons*. Constraint layer B can be divided into three parts: flower ornamental B1, leaf ornamental B2, and garden use potential B3. Criteria layer C consists of 15 indicators, among which flower color, flower type, and flower volume. Since *Rhododendron* in garden application mainly focuses on flower ornamental, the evaluation indexes are mostly flower ornamental in the selection. Finally, the *Rhododendron* is then graded according to the results of the analysis.

To further refine our evaluation, a two-tier aggregation approach was utilized. Initially, AHP assessed individual indicators, grouping them into categories such as flower ornamental, leaf ornamental, and garden use potential. Subsequently, PCA prioritized these categories, identifying the most influential ones for a comprehensive assessment of the *Rhododendron* cultivar’s ornamental value. This integration of PCA and AHP provided a robust framework for our analysis.

In our study, public opinion was indirectly considered through the selection of *Rhododendron* cultivars based on their market acceptance and popularity, as evident in the introduction process described in Section 4.2. However, the primary aim of this research was to elucidate the ornamental value of *Rhododendron*s from a horticultural and botanical perspective. Consequently, the direct survey of public preferences was not the focus, as our intent was to provide a scientifically robust evaluation based on expert knowledge in the field. We acknowledge the value of incorporating a broader range of opinions in future studies, especially from the general public, to complement and enhance the scientific understanding of *Rhododendron*’s ornamental appeal.

#### 4.3.2. Constructing the Judgment Rectangle and One-Time Judgment

Using the established judgment model, *Rhododendron* evaluation criteria were meticulously formulated through an extensive literature review and expert consultations [31]. Professionals were engaged to assess 15 indicators based on their relative significance (see Table 4). Four matrices—A-B, B1-C, B2-C, and B3-C—were generated through rounding and averaging, depicted in Table 3, Table 4, Table 5 and Table 6. The maximum eigenvalue (λmax) and its eigenvector (W) were derived from the calculated matrix for a comprehensive assessment. The consistency ratios (CR) across matrices A-B, B1-C, B2-C, and B3-C, all below 0.1 (refer to Table 5, Table 6, Table 7 and Table 8), validate the validity of the scoring approach [21,25].

As shown in Table 7, the factor weights of flower ornamental B1, leaf ornamental B2, and garden application potential B3 in constraint layer B are 0.687, 0.187, and 0.127, respectively. Among them, flower ornamental B1 accounted for the most, which was significantly larger than the other two factors, indicating that the ornamental of flowers was an important evaluation index of garden comprehensive evaluation. In flower ornamental B2, the comprehensive ranking of flower color C2, flower quantity C5, flower display C6, flower length C9, and flower type C1 was 0.154, 0.103, 0.086, 0.081, 0.077, respectively. Compared with other evaluation indicators, it accounts for a relatively large proportion, so it is the main indicator in comprehensive evaluation. Among them, the weight proportion of flower color C3 and flower quantity C5 was significantly higher than that of other factors, indicating that they played a greater role in comprehensive evaluation. Secondly, factors such as adaptive C15, young leaf C10, mature leaf C11, green stage C12, petal C3, and flower diameter C4 are in the middle position, so they are general evaluation indexes. The remaining factors are secondary evaluation indicators because of the relatively small weight. It can be seen that these results are in line with the actual application of *Rhododendron*s in gardens (Table 9). For example, in the constraint layer B, the flower ornamental B1 weight accounts for the largest proportion, and the flower ornamental B1 is the main judgment basis. This is in line with the actual application of *Rhododendron*s in gardens, which is mainly based on ornamental flowers, and can be used as an ornamental evaluation standard for *Rhododendrons*. 

### 4.4. Statistical Analyses

The data in this study were collated and calculated using Excel 2018, clustering pedigree analysis using SPSS 17.0, heat map drawing using TBtools v2.030., and PCA analysis using the vegan package in R v3.1.2.

## 5. Conclusions

The present study has identified that the proportion of ornamental flowers within the primary index was significantly higher than the other two factors. This underscores the predominant significance of ornamental value when evaluating *Rhododendron* species. Within the secondary indicators, the prevalence of flower color and weight significantly outweighed other factors, exerting the most substantial influence on the evaluation outcomes. Based on the index weights and established scoring criteria, the 24 *Rhododendron* species were categorized into two tiers according to their ornamental value. Notably, *Rhododendron* ‘Xueqing’, *Rhododendron* ‘Biginglian’, and *Rhododendron* ‘Jinyang No. 9’ demonstrated superior ornamental value, indicating their heightened suitability for garden cultivation. 

## Figures and Tables

**Figure 1 plants-13-00558-f001:**
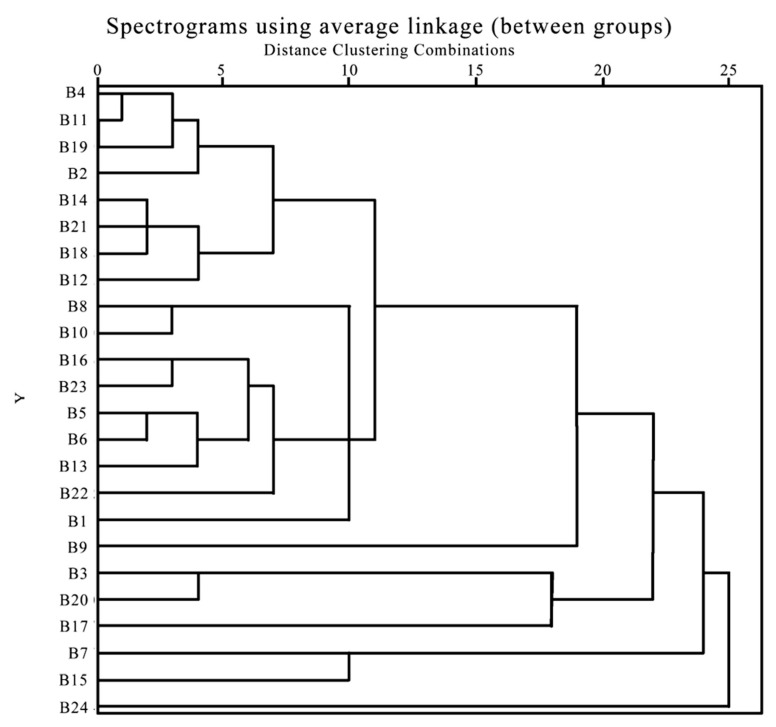
24 *Rhododendron* cultivars score clustering diagram (note: ‘B’ denotes the cultivar serial number).

**Figure 2 plants-13-00558-f002:**
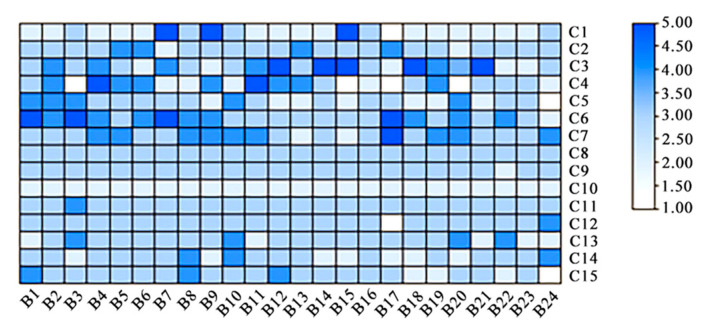
Heat map scores of 24 *Rhododendron* cultivars.

**Figure 3 plants-13-00558-f003:**
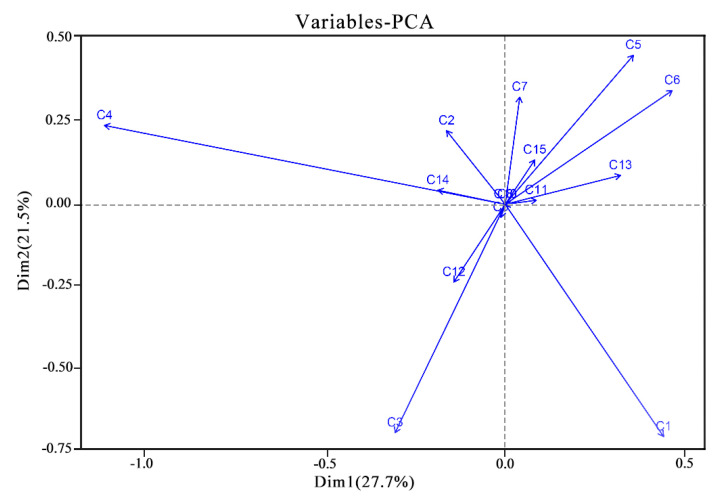
PCA diagram of the evaluation index of 24 *Rhododendron* cultivars.

**Figure 4 plants-13-00558-f004:**
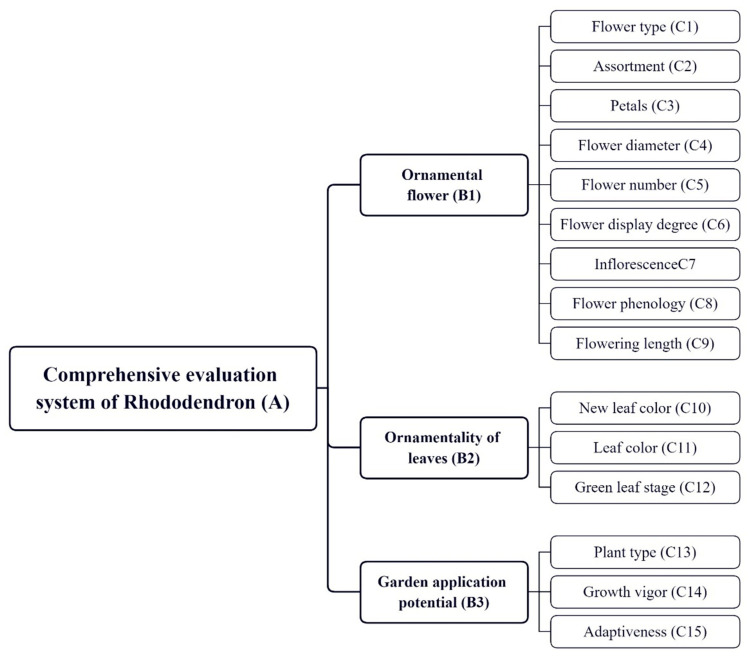
Comprehensive evaluation system model [24].

**Table 1 plants-13-00558-t001:** Evaluation criteria.

Evaluating Indicator	Evaluation Criterion				
	5 Points	4 Points	3 Points	2 Points	1 Points
Flower type C1	Rare color (yellow, green)	Multicolor	Pure color with spotted	Pure color (conventional color)	Pure white
Assortment C2	Bowl-shaped	Funnel-shaped	Full-stacked	Dish-shaped	Horn-shaped
Petal C3	Semi-double petal flowers	Double flower	Petal flowers	Single petal flower	
Flower diameter C4	≥7	6–7	5–6	4–5	3–4
Flower number C5	Great	Large	Relatively large	Relatively sparse	Sparse
Flower display C6	Fully exposed	Most revealed	Partially exposed	Mostly under the flower	Almost all under the flower
Inflorescence C7	>5	4–5	2–3	<2	
Blooming period C8	Multiple a year	Early or late	March to April		
Flowering length C9	60–75 d	40–50 d	30–40 d	25–30 d	20–25 d
New leaf color C10				Yellowish green	
Leaf color C11	Colors	Green	Deep green	Dark green	
Green leaf stage C12		Evergreen, winter foliage discoloration	Evergreen leaves do not change color in winter	Semi-evergreen species	Deciduous species
Plant type C13	Prostrate, tower	Dispersed	Natural Open Type	Sapling	Spherical
Growth vigor C14	Very strong	Strong	Strong general	Weak	Very weak
Adaptiveness C15	Very strong	Strong	Strong general	Weak	Very weak

**Table 2 plants-13-00558-t002:** Comprehensive assessment grade.

Level	Serial Number	Cultivar Name	Aggregate Score
Ⅰ	1	*Rhododendron* ‘Xueqing’	3.167
	2	*Rhododendron* ‘Changchun No. 4’	3.161
	3	*Rhododendron* ‘Wanzi Qianhong’	3.157
	4	*Rhododendron* ‘Shengchun No. 1’	3.152
	5	*Rhododendron* pulchurum ‘YuanYangJin’	3.109
	6	*Rhododendron* ‘Chunbo’	3.108
	7	*Rhododendron* ‘White changchun’	3.103
	8	*Rhododendron* ‘Bing qinglian’	3.095
	9	*Rhododendron* grande Wight	3.085
	10	*Rhododendron* ‘Danzhuang’	3.081
	11	*Rhododendron* ‘Shengchun No. 1’	3.034
	12	*Rhododendron* ‘Jinyang No. 9’	2.984
	13	*Rhododendron* pulchurum’YuDaTaoHua’	2.936
	14	*Rhododendron* ‘Hongyang’	2.933
	15	*Rhododendron* ‘Fengmei No. 1’)	2.908
Ⅱ	16	*Rhododendron* pulchurum ‘Xiaotaohong’	2.879
	17	*Rhododendron* ‘Shanlu’)	2.854
	18	*Rhododendron* ‘Chuixiao’)	2.853
	19	*Rhododendron* ‘Changchun No. 4’	2.841
	20	*Rhododendron* ‘Ruixue’)	2.839
	21	*Rhododendron* ‘Shengchun No. 4’	2.83
	22	*Rhododendron* ‘Huanjing’)	2.783
	23	*Rhododendron* ‘Kunlunyu’	2.777
	24	*Rhododendron* ‘Yinhong Chunjuan’	2.558

**Table 3 plants-13-00558-t003:** 24 *Rhododendron* cultivars.

Number	Cultivar	Assortment	Florescence	Habitat
1	*Rhododendron* ‘Xueqing’	Light grayish-yellow	Spring–summer	Jinhua City, China
2	*Rhododendron* ‘Changchun No. 4’	Strong pink	Spring–summer	Jinhua City, China
3	*Rhododendron* ‘Wanzi Qianhong’	Strong red	Spring–summer	Japan
4	*Rhododendron* ‘Shengchun No. 1’	Slightly desaturated pink	Summer	Jinhua City, China
5	*Rhododendron* pulchurum ‘YuanYangJin’	Light grayish pink and white chimera	Spring–summer	Japan
6	*Rhododendron* ‘Chunbo’	Dark red and light pink	Spring	Jinhua City, China
7	*Rhododendron* ‘White changchun’	White	Spring	Jinhua City, China
8	*Rhododendron* ‘Big qinglian’	White gradient Dark moderate magenta	Spring	Japan
9	*Rhododendron* grande Wight	White or pink	Spring	Taiwan, China
10	*Rhododendron* ‘Danzhuang’	Very soft magenta	Spring	Jinhua City, China
11	*Rhododendron* ‘Shengchun No. 1’	Slightly desaturated magenta	Spring	Jinhua City, China
12	*Rhododendron* ‘Jinyang No. 9’	Pink	Spring	Jinhua City, China
13	*Rhododendron* pulchurum ‘YuDaTaoHua’	Red pink chimera	Spring	Japan
14	*Rhododendron* ‘Hongyang’	Bright pink	Spring	Jinhua City, China
15	*Rhododendron* ‘Fengmei No. 1’	Pink	Spring	Jinhua City, China
16	*Rhododendron* pulchurum ‘Xiaotaohong’	Dark moderate red	Spring	Jinhua City, China
17	*Rhododendron* ‘Shanlu’	Light grayish cyan	Spring	America
18	*Rhododendron* ‘Chuixiao’	Mauve	Spring	Jinhua City, China
19	*Rhododendron* ‘Changchun No. 4’	Soft pink	Spring	Jinhua City, China
20	*Rhododendron* ‘Ruixue’	White	Spring	Jinhua City, China
21	*Rhododendron* ‘Shengchun No. 4’	Slightly desaturated pink	Spring	Jinhua City, China
22	*Rhododendron* ‘Huanjing’	Very soft violet	Spring	Jinhua City, China
23	*Rhododendron* ‘Kunlunyu’	Light grayish pink	Spring	Jinhua City, China
24	*Rhododendron* ‘Yinhong Chunjuan’	Pink with purple tones	Spring	Jinhua City, China

**Table 4 plants-13-00558-t004:** The importance of correlation between factors.

Scale	Implication
1	Where *b* indicates that the two factors are of equal importance.
3	It indicates that one factor is slightly more important than the other factor.
5	It shows that one factor is more important than the other factor.
7	Compared with the two factors, one factor is more important than the other.
9	Compared with the two factors, one factor is more important than the other.
2, 4, 6, 8	Median of the above two adjacent judgments
Reciprocal	If factor i is compared with factor j to judge *b_ij_*, then factor j is compared with factor i to judge *b_ij_*. *b_ji_* = 1/*b_ij_*

**Table 5 plants-13-00558-t005:** The importance of correlation between factors [24].

A	B1	B2	B3	W
B1	1	5	4	0.687
B2	1/5	1	2	0.187
B3	1/4	1/2	1	0.127

Note: λmax = 3.094, CR = 0.0904 < 0.1.

**Table 6 plants-13-00558-t006:** A-B judgment matrix and one-time test [24].

B1	C1	C2	C3	C4	C5	C6	C7	C8	C9	W
C1	1	1/3	1	1	1	2	2	2	1	0.113
C2	2	1	3	2	2	3	3	3	2	0.224
C3	1	1/3	1/2	1/3	1/3	1	2	2	1	0.076
C4	1	1/2	1	1/2	1	1	2	1	1/2	0.087
C5	2	1/2	1	1	2	3	3	2	1	0.15
C6	1	1/2	1	1/2	1	3	3	3	1	0.125
C7	1/2	1/3	1/2	1/3	1/3	1/2	1	1	1/2	0.051
C8	1	1/3	1/2	1/2	1/3	1/2	1	1	1/2	0.058
C9	2	1/2	1	1	1	1	2	2	1	0.118

**Table 7 plants-13-00558-t007:** B2-C judgment matrix and one-time test [24].

B2	C10	C11	C12	W
C10	1	1	1	0.333
C11	1	1	1	0.333
C12	1	1	1	0.333

Note: λmax = 3.000, CR = 0.000 < 0.1.

**Table 8 plants-13-00558-t008:** B3-C judgment matrix and one-time test [24].

B3	C13	C14	C15	W
C13	1	1	1/2	0.250
C14	1	1	1/2	0.250
C15	2	2	1	0.500

Note: λmax = 3.000, CR = 0.000 < 0.1.

**Table 9 plants-13-00558-t009:** Overall ranking values in the comprehensive evaluation system for *Rhododendron* [24].

C	Ornamental Flower B1 0.687	Ornamental Leaves B2 0.187	Garden Application Potential B3 0.127	The Total Ranking Weight of C
Flower type C1	0.113			0.077
Assortment C2	0.224			0.154
Petal C3	0.076			0.052
Flower diameter C4	0.087			0.059
Flower number C5	0.15			0.103
Flower display C6	0.125			0.086
Inflorescence C7	0.051			0.035
Blooming period C8	0.058			0.04
Flowering length C9	0.118			0.081
New leaf color C10		0.333		0.062
Leaf color C11		0.333		0.062
Green leaf stage C12		0.333		0.062
Plant type C13			0.25	0.032
Growth vigor C14			0.25	0.032
Adaptiveness C15			0.50	0.063

## Data Availability

Data is contained within the article and Appendix A.

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
