# Peer review of "Comprehensive Evaluation of Appreciation of Rhododendron Based on Analytic Hierarchy Process"

_plants, 2024, doi:10.3390/plants13040558_

Round 1

Reviewer 1 Report

Comments and Suggestions for Authors

This is an interesting study that classified various plant morphological characteristics of genetic resources of the genus Rhododendron.

However, we recommend supplementing your research with the following:

1. Please provide photos of each class based on morphological characteristics

2. It is recommended to perform phylogenetic analysis by linking molecular biological or chemical characteristics.

​

Comments on the Quality of English Language

Not bad overall

Author Response

I greatly appreciate your valuable suggestions. For detailed responses to your points, please refer to the attached document.

Reviewer 2 Report

Comments and Suggestions for Authors

The paper is interesting because an analytics approach [of sorts] was studied as a means of decision making, presumably to assist planning an ornamental  show garden. This should be of large benefit to landscape planning, especially to novice gardeners or landscape designers. Having said this, the final ranking outcome, presuming I have a correct interpretation of this paper is disappointing. My experience from working with environmental stress physiology of plants, for over forty years, clearly would place adaptation, especially a perennial species, far above other factors. Simply said,  if the cultivar is not well adapted to its' environment, other factors such as flower size, color, presentation, etc., do not matter. If heat stress, cold stress, lack of adaptation to moisture regimes, diseases, pests render the cultivar unworthy, the rest makes no difference.

Accordingly, the scheme proposed may be fine for a specific site that lacks  adaptation limitations, but it would not be useful in most environments, where environmental conditions do limit plant growth. Thus I regard this work to be of limited value, unless the authors can justify otherwise.

Other issues of analytics recommendations are; no controls, no replication, no randomization, thus the application is descriptive and subject to bias from the evaluators.

Comments on the Quality of English Language

Several sentences are written in a passive voice, rewrite to the active form.

Requires  review to be more concise. For example.

L47: Rhododendron cultivars are garden plants with high ornamental value due to their graceful shape, colorful flowers and aromatic odor,  making them one of the ten....

L49: They are widely used....

L51: They are typical indicator plants...

L53: Rhododendrons grown at low altitude areas or low latitudes with higher temperatures are difficult to adapt, and are prone to stunted growth...

L60: Rhododendron cultivars ... [According to international nomenclature guidelines a cultivated taxon with a name is a cultivar, variety is reserved for botanical nomenclature.]

L74: the...

L84: resistance ? ...[to what?]

L110 to l112. Restate in the active not passive voice.

Author Response

(The authors gave the same response as above.)

Reviewer 3 Report

Comments and Suggestions for Authors

Manuscript plants-2720824
Authors: Liang et al.,
Title: Comprehensive Evaluation of Appreciation of Rhododendron Based on Analytic Hierarchy Process

General Comments
       The manuscript has been largely improved since its original version and it is now close to suitability. There are only minor matters that should be looked at.
       However, A main reservation was the exclusion in their surveying of the general opinions. This exclusion should be clearly stated ion them&M and It should indicate the rationale.
       Unclear is also the criteria used in the survey which should also be specified in simply and clear lines.

Specific Comments
       Throughout the manuscript ensure that units and numbers (including temperature and %) are separated by a space.
       Line 75. Remove line “in south-central and southwestern 75 China,” as this reads that Rododendrons are acid indicators only in China, which is not the case.
       Line 116-118 “AHP has been increasingly used to evaluate the ornamental value of garden 116 plants in recent years” change to “AHP has been increasingly used, in recent years, to evaluate the ornamental value of garden plants..”…including Acanthaceae juss …maple….”
       Line 196 “teachers and students” add the number in brackets (n=…)
       Line 201 What criteria were adopted for the 2-tiers aggregation? This is not clear neither in the results nor in the M&M
       Line 217-224. Clearly indicate the 2 obvious clusters (b4 to B12) and B8 to B1 the third is an inconclusive cluster
       Line 224. Add to picture legend what is the letter B
    Line 528 “that decomposes the complex problem” change to “that decomposes a complex problem”

Comments on the Quality of English Language

The manuscript currently still presents small matters which can be resolve by a thorough editing. The reviewer appreciates that with track changes can be problematic, so it is suggested that you remove track changes for editing purposes. 

Author Response

Thank you to the reviewers for their evaluation. Once again, we appreciate your dedication and effort.

Response to Reviewer Comments

General Comments

The manuscript has been largely improved since its original version and it is now close to suitability. There are only minor matters that should be looked at.However, A main reservation was the exclusion in their surveying of the general opinions. This exclusion should be clearly stated ion them&M and It should indicate the rationale. Unclear is also the criteria used in the survey which should also be specified in simply and clear lines.

R:Thank you for your valuable feedback on our manuscript. In response to your concern regarding the consideration of public opinion in our study, we wish to clarify the following, as now detailed in lines 312-320 of our revised manuscript:

"In our study, public opinion was indirectly considered through the selection of Rhododendron cultivars based on their market acceptance and popularity. This selection process, as described in Section 4.2, reflects the indirect assessment of public preferences through existing market trends. However, the primary aim of this research was to elucidate the ornamental value of Rhododendrons from a horticultural and botanical perspective, focusing on a scientific evaluation rather than a direct public survey. This approach was chosen to ensure a scientifically robust evaluation based on expert knowledge. Nonetheless, we acknowledge the importance of including a broader range of opinions, particularly from the general public, in future studies to further enrich our understanding of Rhododendron's ornamental appeal."

We believe this addition adequately addresses your concern and enhances the manuscript by clarifying our methodology and intent. We appreciate your suggestion and look forward to further guidance.

Detailed Comments

  1. Throughout the manuscript ensure that units and numbers (including temperature and %) are separated by a space.

R:Thank you for your suggestion on formatting units and numbers in our manuscript, including spaces for temperature, percentages, and citations. We have made the necessary revisions.         

  1. Line 75. Remove line “in south-central and southwestern 75 China,” as this reads that Rododendrons are acid indicators only in China, which is not the case.

R: Thank you for your insightful comment regarding the description of Rhododendrons in our manuscript. In line with your suggestion, we have revised the text to provide a more accurate and global perspective on the habitat preferences of Rhododendrons.

In the revised manuscript (now reflected in line 56-57), the text has been modified as follows:

"They are widely recognized as typical indicators of acidic soils, thriving in humid and cold regions with diffused light, and displaying intolerance to heat."

  1. Line 116-118 “AHP has been increasingly used to evaluate the ornamental value of garden 116 plants in recent years” change to “AHP has been increasingly used, in recent years, to evaluate the ornamental value of garden plants..”…including Acanthaceae juss …maple….”

R: Thank you for your suggestion. We have replaced the sentence with "AHP has been increasingly used, in recent years, to evaluate the ornamental value of garden plants...including Acanthaceae juss...maple..." as you advised.

  1. Line 196 “teachers and students” add the number in brackets (n=…)

R: Thank you for your valuable suggestion. We have now added the specific number of participants in brackets (n=35) following "teachers and students" , as you recommended.

  1. Line 201 What criteria were adopted for the 2-tiers aggregation? This is not clear neither in the results nor in the M&M

R: Thank you for pointing out the need for clarity regarding the criteria adopted for the two-tier aggregation in our study.

In response to your query, we have added a detailed explanation of the two-tier aggregation process in Section 4.3.1 of our revised manuscript. This addition specifically outlines how we initially used the Analytic Hierarchy Process (AHP) to assess individual indicators, categorizing them into broader groups such as flower ornamental, leaf ornamental, and garden use potential. Following this, we applied Principal Component Analysis (PCA) to these categories to prioritize and identify the most significant ones, thereby enhancing the comprehensiveness of our evaluation framework.

We believe these additions provide the necessary clarity on the criteria used in our two-tier aggregation approach and hope that this adequately addresses your concern.

  1. Line 217-224. Clearly indicate the 2 obvious clusters (b4 to B12) and B8 to B1 the third is an inconclusive cluster

R:Thank you for your insightful comments. We have revised the manuscript to clearly delineate the clusters within the Rhododendron cultivars as you suggested. Specifically, we have now explicitly identified and described the two obvious clusters (from B4 to B12 and B8 to B1), and have acknowledged the third as less defined, indicating a need for further investigation.

  1. Line 224. Add to picture legend what is the letter B

R:Thank you for your suggestion. We have added a note to the legend of Figure 1 to clarify that 'B' denotes the cultivar serial number.

  1. Line 528 “that decomposes the complex problem” change to “that decomposes a complex problem”

R:Thank you for your recommendation. The sentence on line 528 has been modified to "that decomposes a complex problem" as suggested.

9.The manuscript currently still presents small matters which can be resolve by a thorough editing. The reviewer appreciates that with track changes can be problematic, so it is suggested that you remove track changes for editing purposes.

R:Thank you for your feedback. We have conducted a thorough review of the manuscript to address the minor English language issues. Additionally, in line with your suggestion, we have stopped using track changes mode. Instead, we have highlighted the major revisions for ease of identification and review.

Round 2

Reviewer 1 Report

Comments and Suggestions for Authors

I have confirmed that the quality of your paper has improved. Unfortunately, I can no longer comment on this paper.

Author Response

Dear reviewer:

We sincerely appreciate your valuable contribution during the review process. Although we did not receive further suggestion from your side this time, your previous assessment and established grading criteria were instrumental in guiding us in making significant revisions to the manuscript.

Attached to this letter is a revised version of our manuscript. We strive to fully incorporate your insights and suggestions with the aim of improving the quality and rigor of our work.

We greatly appreciate the time and effort you expended in reviewing our paper and we believe these revisions significantly improved the manuscript. We hope that the changes we make accurately reflect your feedback and comply with the journal's standards.

Thank you again for your valuable comments and guidance.
